# Prognostic Value of Hematological Parameters in Oral Squamous Cell Carcinoma

**DOI:** 10.3390/cancers15215245

**Published:** 2023-10-31

**Authors:** Lorenzo Fernandes Moça Trevisani, Isabelle Fernandes Kulcsar, Marco Aurélio Vamondes Kulcsar, Rogerio Aparecido Dedivitis, Luiz Paulo Kowalski, Leandro Luongo Matos

**Affiliations:** 1Programa de Pós-Graduação em Anestesiologia, Ciências Cirúrgicas e Medicina Perioperatória, Faculdade de Medicina, Universidade de São Paulo, São Paulo 01246-903, Brazil; lorenzofmtrevisani@hc.fm.usp.br; 2Instituto do Câncer do Estado de São Paulo (Icesp), Faculdade de Medicina, Universidade de São Paulo, São Paulo 01246-000, Brazil; isabelle.kulcsar@hc.fm.usp.br; 3Head and Neck Surgery Department, Instituto do Câncer do Estado de São Paulo (Icesp), Faculdade de Medicina, Universidade de São Paulo, São Paulo 01246-000, Brazil; marco.kulcsar@hc.fm.usp.br; 4Head and Neck Surgery Department, Hospital das Clínicas, Faculdade de Medicina, Universidade de São Paulo, São Paulo 01246-000, Brazil; r.dedivitis@fm.usp.br; 5Head and Neck Surgery Department, Faculdade de Medicina, Universidade de São Paulo, São Paulo 01246-000, Brazil; luiz.kowalski@hc.fm.usp.br; 6Faculdade Israelita de Ciências da Saúde Albert Einstein, São Paulo 05652-000, Brazil

**Keywords:** oral squamous cell carcinoma, prognosis, RDW

## Abstract

**Simple Summary:**

Oral squamous cell carcinoma is a challenging mouth cancer, with roughly half of those diagnosed succumbing to it, often due to it recurring or spreading to other body parts. Although some factors like tumor size, invasiveness, lymph node involvement, and specific characteristics when examining tissue samples are known to influence the disease’s severity, the role of blood-based parameters in predicting outcomes is less clear. In our research, we discovered that an easily measured blood parameter, known as RDW (red cell distribution width), exceeding 14.3%, can be just as effective as traditional factors in forecasting a patient’s risk of dying from this cancer. If a patient’s RDW is high, it is a signal for doctors to closely monitor and have more in-depth treatment discussions with the patient before initiating any therapies. This blood test offers a simple and accessible way to improve patient care and decision-making.

**Abstract:**

Introduction: Oral squamous cell carcinoma (OSCC) remains a significant public health concern. The variables utilized to determine appropriate treatment for this disease also represent its most unfavorable prognostic factors, with these parameters solely determined by the neoplasm and its behavior. However, a lack of well-established indices is evident in the literature that specifically relate to the patient and indicate a worse prognosis. Objective: To assess the prognostic impact of hematological indices in patients with OSCC. Methods: This retrospective cohort study included patients with oral squamous cell carcinoma (OSCC) who underwent curative-intent treatment. Treatment encompassed surgery, followed by adjuvant therapy, as necessary. Laboratory tests were conducted immediately prior to surgery, and demographic information was obtained from medical records. Results: The cohort comprised 600 patients, with 73.5% being male subjects. Adjuvant treatment was recommended for 60.3% of patients. Throughout the follow-up period, 48.8% of participants died. Univariate analysis indicated that perineural invasion, angiolymphatic invasion, pT4 tumors, lymph node metastases, extranodal extravasation, RDW > 14.3%, NLR (neutrophil–lymphocyte ratio) > 3.38, PLR (platelet–lymphocyte ratio) > 167.3, and SII (systemic inflammatory/immune response index) > 416.1 were factors associated with increased mortality. These threshold values were established through ROC curve analysis. In the multivariate analysis, angiolymphatic invasion (HR = 1.43; 95% CI: 1.076–1.925; *p* = 0.014), pT4a/b tumors (HR = 1.761; 95% CI: 1.327–2.337; *p* < 0.001), extranodal extravasation (HR = 1.420; 95% CI: 1.047–1.926; *p* = 0.024), and RDW (HR = 1.541; 95% CI: 1.153–2.056; *p* = 0.003) were identified as independent risk factors for decreased overall survival. Conclusions: RDW > 14.3% was proven to be a reliable parameter for assessing overall survival in patients with OSCC. Further studies are required to evaluate the clinical applicability of other hematological indices.

## 1. Introduction

Oral squamous cell carcinoma (OSCC) is still a public health problem. According to data published by the International Agency for Research on Cancer (IARC), it is the eighth most common malignant neoplasm among men worldwide. Considering only countries with medium and low human development indexes, it is the third most common in males and the sixth most lethal [1]. In Brazil, according to data from the National Cancer Institute, it is the fifth most common cancer among men and the thirteenth among women, with 6295 deaths identified from this disease in the country in 2017 [2]. Its diagnosis and treatment are often performed late when metastatic lesions are advanced, a fact that worsens the prognosis, increases the difficulty in treatment and leads to aesthetic and functional changes [3,4].

The study of prognosis in cancer patients is of the greatest importance. Since, with knowledge of the variables affecting the disease outcome, it is possible to better understand the biological behavior of the tumor and the natural history of the disease [5]. Thus, it is possible to promote an individualized treatment for each patient, with more intensified options available in tumors with worse or milder prognoses, if a better response to treatment is expected [6]. In addition, the disease outcome is an issue frequently raised by patients with cancer and their families; therefore, the more data available, the more accurate the information provided by the assistant physician [7].

In oral cancer, different factors influence the disease outcome [8]. The main ones are covered by the TNM staging of the American Joint Committee on Cancer (AJCC), which is currently in its eighth edition [9], advanced stage of the primary tumor and the presence of lymph node or distant metastases continue to be the main factors that negatively influence the disease prognosis [10]. The biological behavior and the microenvironment of the tumor are also of great importance in this matter. Lesions developing with perineural or angiolymphatic invasion also have worse prognoses [11,12,13,14,15]. Other variables related to the patient, such as age, comorbidities, and nutritional status, can also directly influence the prognosis [10,12,16,17,18,19].

Recently, some parameters connected to the systemic inflammatory response have been associated with a worse prognosis for OSCC. The tumor microenvironment contains the most varied immune cells, such as macrophages, neutrophils, dendritic cells, natural killer, and T- and B-lymphocytes, composing tumor inflammation and the antitumor immune response, which dictate the balance between the environment and the progression of the lesion [20]. Many of these data can be obtained simply by evaluating the pretreatment blood count; parameters that are quite simple in clinical practice.

The preoperative dosage of inflammatory components and their relationship to each other, through the creation of hematological indices, has been increasingly related to a worse prognosis in cancer patients. Peritumoral lymphocytic infiltrate has been associated with benefits in cancer survival [21]. Most of the inflammatory indices studied use the serum dosage of lymphocytes, which are the neutrophil–lymphocyte ratio (NLR), platelet–lymphocyte ratio (PLR), lymphocyte–monocyte ratio (LMR), and the systemic immune/inflammatory (SII) response index, which is calculated through multiplying platelets by neutrophils and dividing by lymphocytes. Previous studies have linked these indices to a worse prognosis in head and neck cancer [22,23,24,25]. Likewise, the relationship between increased RDW (red blood cell distribution width) and an unfavorable outcome in cancer patients has also been studied. It has already been shown that an increased RDW can be associated with an unfavorable prognosis in different malignant neoplasms. This fact is attributed to possible oxidative stress, inflammation, and malnutrition [26]. It was also identified that anemia can have a negative impact on the evolution of malignant neoplasia [27] and another newly studied index is calculated by the hemoglobin (Hb) to RDW (RHbRDW) ratio. Likewise, an association between this index and worse disease outcomes was demonstrated in esophageal and head and neck neoplasms [28,29].

Based on that, the objective of this study was to evaluate the impact of different hematological indices on the prognosis of patients with oral squamous cell carcinoma.

## 2. Methods

This was a retrospective cohort study approved by our Institutional Review Board (CAAE: 32884214.5.0000.0065). Informed consent was obtained from all live subjects involved in the study. Patient consent was waived for deceased patients and those not localized to sign the document. All consecutive patients aged 18 years or older and surgically treated with curative intent for OSCC at the Instituto do Câncer do Estado de São Paulo, Hospital das Clínicas da Faculdade de Medicina da Universidade de São Paulo (ICESP HCFMUSP) from October 2009 to December 2018 were included. Patients with other previous oncological treatments, those with recurrent disease or with distant metastasis at presentation (M1), and those with SCC of the lip were not included.

All patients underwent surgical resection of the primary tumor associated with a cervical lymph node approach and adjuvant therapy when indicated. Patients with cervical metastasis diagnosed either preoperatively or intraoperatively underwent radical or modified radical neck dissection, which consists of the resection of cervical levels I to V, which represents the submandibular, internal jugular, and supraclavicular lymph node chains ipsilateral to the metastasis. Patients with no evidence of cervical metastasis received an elective neck dissection (levels I–III) and selected T1 or T2 cases classified by the eighth edition of the AJCC received sentinel lymph node biopsy, followed by radical neck dissection, and adjuvant therapy, in cases of positive pNs. Adjuvant radiotherapy was indicated in patients with cervical metastases or with perineural or angiolymphatic invasion, or further, in advanced primary tumors, classified as pT4 and some selected cases of pT3. Finally, patients with positive or close margins after resection or extranodal extension of lymph node metastases received cisplatin-based chemotherapy concomitant with radiotherapy.

Pathological specimens were analyzed by a single team from the institution. Disease staging was based on the eighth edition of the AJCC. Patients undergoing treatment prior to the publication of the eighth edition had their staging revised for the updated edition through a review of the anatomopathological reports. Clinical and demographic data were obtained by electronic medical record review.

Hematological indices were obtained from the latest laboratory exams, which had been collected before the surgery. Patients were followed monthly for the first year, bimonthly for the second, and biannually in subsequent years. Follow-up time was calculated from the first day of treatment to the date of death or the date of the last medical appointment. 

### Statistical Analysis

The values obtained from the quantitative variables were organized and described using mean and standard deviation (SD), as well as the minimum and maximum values. Absolute and relative frequencies were used for qualitative data. ROC (receiver operator characteristic curve) was used to identify the ideal cutoff value for the studied indexes, using death from any cause as the outcome. The Cox proportional regression model was used in both univariate and multivariate analyses, estimating the hazard ratio (HR) and 95% confidence interval (95% CI) values. Variables with a *p*-value < 0.10 were included in the multivariate analysis. A Kaplan–Meier curve was used in the survival analyses, and the log-rank test was applied to compare the curves. A *p*-value of 5% or less (*p* ≤ 0.05) was adopted as the probability of error α or I. All statistical tests were performed using SPSS 27.0 software (IBM^®^ Inc.; Endicott, NY, USA).

## 3. Results

The study cohort consisted of 600 patients, formed predominantly of males (73.5%). Most patients were smokers (78.2%) who also had a history of alcohol abuse (65.2%). The oral cavity subsites most affected by the disease were the tongue, the floor of the mouth, and the retromolar region, which represented 40%, 27.3%, and 11.2% of the cases, respectively. A total of 40.8% of the patients had tumors classified as pT4, and 50.2% had lymph node metastases at the time of surgical treatment.

From the studied cohort, 362 (60.3%) patients received adjuvant radiotherapy and of these, 135 were treated concomitantly with chemotherapy. During the follow-up, 25.7% of the cases exhibited disease recurrence, 11.3% had distant metastases, and 48.8% died. The mean follow-up time was 33.1 months. Full descriptive data are detailed in Table 1.

As for the studied preoperative hematological indexes, RDW ranged from 11.3% to 26.6%, with an average of 13% (SD:2%), while the optimized cutoff for overall survival (OS) in this cohort was 14.3%. The NLR ranged from 0.3 to 26.9, with an average of 3.2, while the optimized cutoff value was 3.3. The PLR had an average of 150.9 and an optimized cutoff value of 167.3. The SII presented an average of 884 and a cutoff value of 416.1. The average of the LMR and RHbRDW indexes was 4.4 and 1, respectively. In the study of the ROC curve to identify the optimized cutoff values for overall survival, the LMR and RHbRDW showed an area under the curve below 0.5, which characterizes them as inappropriate indexes for the assessment of overall survival in this case-by-case evaluation. Based on that, these two indexes were not further applied in the evaluation of the patient’s disease outcomes.

Univariate analysis showed that perineural invasion, angiolymphatic invasion, pT4a/b tumors, lymph node metastases, extranodal extension, RDW > 14.3%, NLR ratio > 3.38, PLR > 167.3, and SII > 416,1 were all factors associated with higher mortality, as shown in Table 2.

As the NLR, PLR, and SII use the serum lymphocyte count, a multivariate analysis was performed only among these three indexes to assess the dependence of these parameters on each other (Table 3). Among them, the PLR > 167.3 was shown to be a factor independently related to overall survival (HR = 1.37; 95% CI: 1.029–1.824; *p* = 0.031). Therefore, only PLR was maintained in the final multivariate analysis.

Multivariate analysis identified angiolymphatic invasion as independent risk factors for lower overall survival (HR = 1.43; 95% CI: 1.076–1.925; *p* = 0.014), pT4a/b tumors (HR = 1.761; 95% CI: 1.327–2.337; *p* < 0.001), extranodal extension (HR = 1.420; 95% CI: 1.047–1.926; *p* = 0.024), and the RDW (HR = 1.541; 95% CI: 1.153–2.056; *p* = 0.003), as detailed in Table 4. Patients with RDW ≤ 14.3% had a median survival of only 55 months, whereas in the RDW group > 14.3%, this only reached 26 months. The other data on median survivals and cumulative survivals are described in Table 5. Cumulative survivals of all independent variables were demonstrated by Kaplan–Meier curves (Figure 1).

## 4. Discussion

The evaluation and identification of systemic variables that may reflect a worse prognosis in patients with different diseases have been increasingly studied [30]. To date, our study represents the most extensive analysis in the existing literature, encompassing an evaluation of six distinct systemic indices associated with an unfavorable prognosis among patients with OSCC. We also identified that preoperative RDW > 14.3% was an independent risk factor for lower overall survival. RDW was similar to factors already well established in the literature, which were also independent variables for lower OS, such as angiolymphatic invasion, extranodal extravasation, and advanced primary tumors (pT4).

RDW was identified for the first time as a worse prognostic factor in 2007, through an analysis performed on an American database that gathered 4809 patients with cardiac events. In this series, 36 hematological variables were evaluated, and increased RDW was identified as an independent factor for adverse events and higher mortality [31]. After this first study, RDW elevation was investigated for multiple chronic disorders, such as venous thromboembolism, diabetes, chronic kidney disease, liver dysfunction, and chronic obstructive pulmonary disease, and it was observed that RDW elevation may be related to worse prognosis [32,33,34,35,36]. Indeed, it has been demonstrated that elevated RDW is correlated with higher mortality across the general population. A cohort study of 15,852 patients showed that patients in the highest quintile of the RDW had an increased risk of mortality of 27%, compared to the lowest quintile [37].

Regarding cancer patients, RDW has been studied in two main aspects. The first considers the aid in the differential diagnosis in patients with suspicious lesions for malignancy, and the second in the prognostic evaluation of patients already diagnosed with malignant neoplasms. RDW has already been shown to be useful in the differential diagnosis of malignant and benign lesions in breast tumors [38], bile duct obstruction [39], endometrial bleeding [39], and colonic lesions [40]. When high, it indicated a higher probability of malignant neoplasm.

Considering the assessment of prognosis and mortality, RDW has already been studied in different primary tumors. In the literature, it is possible to find two systematic reviews with meta-analyses on the topic. The first, published in 2017, highlights 16 studies and covers 4267 patients. High RDW was shown to be associated with worse overall survival (HR = 1.47; 95% CI: 1.29–1.66) and worse cancer-specific survival (HR = 1.46; 95% CI: 1.08–1.85) for different types of malignant neoplasms [41]. The second published study gathered 49 articles relating the prognosis of cancer patients to RDW. Studies were analyzed for 16 different types of solid malignant tumors and five types of hematologic neoplasms. The solid tumors that showed the greatest relation to RDW were colorectal carcinoma, hepatocarcinoma, and non-small cell lung cancer, whereas breast and urothelial tract cancer did not show a clear relationship in this review [26]. These data corroborate the results of the present study, which also found a relationship between RDW and lower overall survival for OSCC. Furthermore, the association between the increase in RDW and the decrease in overall survival was evidenced in 45 of the 49 analyzed studies, while 26 also showed a relationship between progression-free survival, disease-free survival, and/or recurrence-free survival [26].

In relation to head and neck cancers, in the last 5 years, there have been some studies that have evaluated the application of RDW in the disease outcome. The first series on the topic investigated prognostic factors in 103 patients with laryngeal squamous cell carcinoma. Among all the variables studied, only RDW and TNM staging were independent factors for increased mortality [42]. After this first study, other authors found similar results in case series of patients with laryngeal SCC [43,44]. In another study, 177 patients with laryngeal SCC were retrospectively evaluated, and increased RDW was related to non-specific complications of laryngectomy, such as deep vein thrombosis, pneumonia, cardiac events, and difficulty in weaning from ventilation, although not with decreased overall survival [45]. Regarding primary pharyngeal tumors, a retrospective study that analyzed 2318 patients with nasopharyngeal carcinoma for a minimum period of 4 years found that RDW alone was not an independent factor for higher mortality but only when it associated with body mass index—BMI (RDW > 13.55% and BMI < 18.5). This score proved to be an independent factor for decreased overall survival [46]. As for oropharyngeal SCC, a study published in 2021 retrospectively analyzed 208 cases and patients with RDW above 13.8% showed a decrease in overall survival in the univariate analysis; however, this result did not remain an independent factor in the multivariate analysis [47].

There are few studies in the literature that relate the prognosis of OSCC patients to RDW. The first series published in 2017 included 374 OSCC patients and found no difference in disease recurrence or overall survival among patients with increased RDW (cutoff: > 14.5%) [48]. In another publication, patients with RDW ≥ 15% were more likely to have lymph node metastases, advanced staging, and greater primary tumor volume, while this same group also had a lower overall survival in the multivariate analysis (HR = 1.46; 95% CI: 1.13–2.86) [49]. The cutoff value in this study was stipulated based on the service laboratory’s upper reference limit and was not statistically studied. Another recent publication included 74 patients with oral tongue and base of the tongue SCC. The RDW cutoff value was 13.5%, calculated by the ROC curve, and the 5-year overall survival was 67% and 26% in patients with RDW < 13.5% and RDW ≥ 13.5%, respectively. Between these two groups, there was no difference in terms of age, sex, staging, tumor volume, lymph node metastases, and comorbidities, among other variables. The multivariate analysis in this study showed an important relationship between RDW values ≥ 13.5% and lower overall survival at 5 years [50]. In a recent publication, our group also showed that RDW can also be used as an index of lower overall survival in patients with pT4 OSCC [19].

The adopted cutoff values were determined in different ways among the different studies published in the literature. The main one is the calculation by the ROC curve, although reference values predetermined by the laboratories of the institutions were also used, as well as the adoption of cutoff values identified in previous studies [26]. In most of the studies included in the two systematic reviews available in the literature, the cutoff values varied between 12.2% and 20% [26,41], with 80% of the cutoff values being between 13% and 15% [26]. In the present study, the ideal cutoff value identified by the ROC curve for assessing overall survival was 14.3%, also in accordance with what was previously identified in another publication by our group [19]. We did not study the cutoff value using a time-dependent method since the applications of these methodologies in clinical studies are still lacking [51].

The pathophysiological link of increased RDW with chronic morbidities and worse prognosis is not yet fully understood. It is known that the action of erythropoietin (EPO) in the bone marrow is a determinant in the alteration of RDW [50]. Proinflammatory cytokines have the potential to inhibit the synthesis and activity of EPO [52], meaning there may be a direct relationship between chronic inflammatory states and increased RDW. In addition, inflammation can impair iron metabolism by interfering with erythropoiesis [53] and, ultimately, can decrease the half-life of red blood cells, requiring the release of immature erythrocytes into the bloodstream [54]. Oxidative stress can also increase RDW by having an important influence on erythrocyte half-life. Likewise, chronic destruction states with iron, folic acid, and vitamin B12 deficiency can also lead to anisocytosis [54].

However, the real reason there is an increase in RDW in patients with a worse prognosis is still unclear. New research lines have associated the increase in RDW with acute and chronic hypoxemic conditions. Hypoxemia stimulates peaks of EPO release, and this increases not only the production of erythrocytes but also their size. Thus, frames that decrease oximetry may increase anisocytosis [55]. In our cohort, patients with high RDW had a shorter survival time and it can be hypothesized that early mortality in the high RDW group may be related to the unfavorable condition of the host and not only to the pathological characteristics of the neoplasm itself.

As it is a retrospective cohort, this study has limitations. The indices considering the serum count of lymphocytes NLR, PLR, and SII were identified in the univariate analysis as indicators for the lowest overall survival, although this result was not found in the multivariate analysis. Previous studies have already shown that these indexes are associated with a worse outcome for OSCC [48,56,57], and our study represents one more. However, even with a robust cohort of patients and evaluating different hematological parameters, it could be observed as a limitation and also a virtue of the study. Further investigation is still needed to identify why these indexes were not effective in demonstrating lower overall survival similar to in the current study. A hypothesis that should be considered is that, in relation to other series, our case series has a high number of pT4 tumors (40.8%) and consequently higher mortality rates, which could mask the effectiveness of the studied indexes and perhaps provide less statistical impact. The study does not provide a validation analysis using an external cohort of patients, which limits the calibration of the predictive results, and it was not possible to build a nomogram, for example, a significant limitation that should be addressed.

## 5. Conclusions

In the present study, an important association was seen between increased RDW and higher mortality in a robust cohort of OSCC patients initially treated by surgery. This fact consolidates and expands the applicability of this worse prognosis index. More studies are needed to determine an optimal cutoff value, to understand the biological mechanisms behind the findings, and to prospectively assess their clinical applicability.

## Figures and Tables

**Figure 1 cancers-15-05245-f001:**
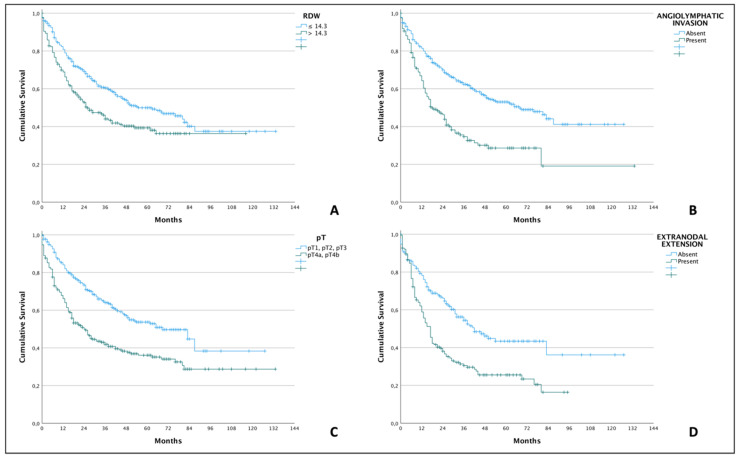
Kaplan–Meier curves demonstrating lower overall survival (*p* < 0.001; log-rank test for all comparisons) in OSCC patients with RDW > 14.3% (**A**), presence of angiolymphatic invasion (**B**), pT4-stage (**C**), and presence of extranodal extension of cervical metastases (**D**).

**Table 1 cancers-15-05245-t001:** Descriptive data from the study cohort.

Variable	Results
Patient’s/Tumor’s characteristics	
Male	441 (73.5%)
Age (years old)	61.3 ± 11.7 (15–91) *
Subsites of primary tumor	
Tongue	240 (40%)
Floor of the mouth	164 (27.3%)
Retromolar area	67 (11.2%)
Inferior gum	55 (9.2%)
Buccal mucosa	42 (7%)
Superior gum	19 (3.2%)
Hard palate	13 (2.2%)
Smoking	469 (78.2%)
Alcohol abuse	391 (65.2%)
Histopathological data	
Greatest tumor dimension (cm) *	3.54 ± 1.8 (0.1–11) *
Deep of invasion (cm) *	1.8 ± 1.4 (0.1–7.9) *
Degree of differentiation	
Well	157 (26.2%)
Moderate	379 (63.2%)
Low	54 (9%)
Perineural invasion	302 (50.3%)
Angiolymphatic invasion	160 (26.7%)
Positive margins	75 (12.5%)
pT classification	
pT1	132 (22%)
pT2	140 (23.3%)
pT3	79 (13.2%)
pT4a/b	245 (40.8%)
Lymph node metastasis	301 (50.2%)
pN classification	
pN0	299 (49.8%)
pN1	51 (8.5%)
pN2a	24 (4%)
pN2b	53 (8.8%)
pN2c	27 (4.5%)
pN3b	146 (24.3%)
Extranodal extension	165 (46.3%)
Clinical and follow-up data	
RDW (%)	13 ± 2.0 (11.3–26.6%) *
NLR	3.22 ± 2.7 (0.3–26.95) *
PLR	150.9 ± 88.2 (34.8–1141.8) *
LMR	4.4 ± 9.5 (0.38–185) *
SII	884 ± 920.7 (27.7–13,234.1) *
RHbRDW	1 ± 0.7 (0.3–10.1) *
Adjuvant radiotherapy	362 (60.3%)
Adjuvant chemotherapy	135 (22.5%)
Locoregional recurrence	154 (25.7%)
Distant metastasis	68 (11.3%)
Death	293 (48.8%)
Follow-up (months)	33.1 ± 27 (0–133)

***** Mean ± standard deviation (minimum–maximum). Legend: RDW—red blood cell distribution width; NLR—neutrophil–lymphocyte ratio; PLR—platelet–lymphocyte ratio; LMR—lymphocyte–monocyte ratio; SII systemic immune/inflammatory response index calculated by multiplying platelets by neutrophils and divided by lymphocytes; RHbRDW—hemoglobin to RDW ratio.

**Table 2 cancers-15-05245-t002:** Univariate analysis of factors associated with lower overall survival in OSCC patients.

Variable	HR	95% CI	*p*-Value *
Male	1.010	0.780–1.316	0.923
Smoking	0.942	0.709–1.251	0.678
Alcohol abuse	1.032	0.808–1.318	0.802
Positive margins	1.350	0.976–1.867	0.070
Low differentiation	1.038	0.689–1.566	0.858
Perineural invasion	2.100	1.655–2.663	<0.001
Angiolymphatic invasion	2.081	1.632–2.653	<0.001
pT4a/b	1.866	1.482–2.350	<0.001
Lymph node metastasis	2.443	1.916–3.115	<0.001
Extranodal extension	1.888	1.430–2.493	<0.001
RDW > 14.3	1.509	1.191–1.910	<0.001
NLR > 3.38	1.568	1.231–1.998	<0.001
PLR > 167.3	1.585	1.250–2.010	<0.001
SII > 416.1	1.525	1.159–2.007	0.003

* Cox regression model (univariate). Legend: HR—hazard ratio; 95% CI—95% confidence interval; RDW—red blood cell distribution width; NLR—neutrophil–lymphocyte ratio; PLR—platelet–lymphocyte ratio; SII systemic immune/inflammatory response index calculated by multiplying platelets by neutrophils and divided by lymphocytes.

**Table 3 cancers-15-05245-t003:** Multivariate analysis of lymphocyte-linked indexes.

Variable	HR	95% CI	*p*-Value *
NLR > 3.38	1.315	0.983–1.761	0.065
PLR > 167.3	1.370	1.029–1.824	0.031
SII > 416.1	1.255	0.924–1.704	0.145

* Cox regression model (multivariate). Legend: HR—hazard ratio; 95% CI—95% confidence interval; NLR—neutrophil–lymphocyte ratio; PLR—platelet–lymphocyte ratio; SII systemic immune/inflammatory response index calculated by multiplying platelets by neutrophils and divided by lymphocytes.

**Table 4 cancers-15-05245-t004:** Multivariate analysis of independent risk factors for lower overall survival.

Variable	HR	95% CI	*p*-Value *
Perineural invasion	1.291	0.30–1.793	0.128
Angiolymphatic invasion	1.439	1.076–1.925	0.014
pT4a/b	1.761	1.327–2.337	<0.001
Lymph node metastasis	1.654	0.911–3.004	0.098
Extranodal extension	1.420	1.047–1.926	0.024
RDW > 14.3	1.541	1.156–2.056	0.003
PLR >167.3	1.125	0.838–1.510	0.432

* Cox regression model (multivariate). Legend: HR—hazard ratio; 95% CI—95% confidence interval; RDW—red blood cell distribution width; PLR—platelet–lymphocyte ratio.

**Table 5 cancers-15-05245-t005:** Survival analysis according to independent risk factors identified by multivariate analysis.

Variable		Events/Total	Median Survival	Cumulative Survival (140 Months)	*p*-Value *
RDW	≤14.3%	179/400	55 months	37.5%	<0.001
	>14.3%	113/199	26 months	36.3%	
ALI	Negative	188/436	68 months	41.2%	<0.001
	Positive	103/160	18 months	19.1%	
pT	pT1 pT2 pT3	143/351	69 months	38.3%	<0.001
	pT4a/b	148/244	24 months	28.7%	
ENE	Negative	88/191	42 months	36.2%	<0.001
	Positive	116/165	17 months	16.4%	

* Log-rank test. Legend: RDW—red blood cell distribution width; ALI—angiolymphatic invasion; ENE—extranodal extension.

## Data Availability

Raw data may be made available by the corresponding author upon reasonable request.

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
