# Peer review of "Prognostic Value of Hematological Parameters in Oral Squamous Cell Carcinoma"

_cancers, 2023, doi:10.3390/cancers15215245_

Round 1

Reviewer 1 Report

Comments and Suggestions for Authors

Thanks for giving me the chance to review this well written manuscript. The authors explored prognostic value of hematological parameters in OSCC with 600 patients. I have some concerns about this manuscript.

1. This manuscript lacks of novelty.

2. The key words should not includ the "Nutrophils; Lymphocytes; Monocytes; Platelets" ?

3. ROC curve is not suited for time-related endpoints included the OS.

4. Detailed inclusion and exclusion criteria are needed.

Comments on the Quality of English Language

Minor editing

Author Response

Answers to Reviewer 1:

General comments: Thanks for giving me the chance to review this well written manuscript. The authors explored prognostic value of hematological parameters in OSCC with 600 patients. I have some concerns about this manuscript.

  1. This manuscript lacks of novelty.

Answer: Thank you for your revision. As described at the discussion, to date, our study represents the most extensive analysis in the existing literature, encompassing an evaluation of six distinct systemic indices associated with an unfavorable prognosis among patients with OSCC. We know that the main result could not be specifically new, but our cohort is robust, and the data could also contribute with the literature once this kind of evaluation is scarce. We highlighted this point at the text and also added this point as a limitation at the end of the paper.

  1. The key words should not includ the "Nutrophils; Lymphocytes; Monocytes; Platelets"?

Answer: These keywords were excluded from the manuscript as requested.

  1. ROC curve is not suited for time-related endpoints included the OS.

Answer: Thank you again for this observation. We completely understand that, however there is no standard method to determine cutoff values in time-depending endpoints. Moreover, the calculated endpoint in our study was similar to other previous reports, as demonstrated in line 286 (page 9). In a study published in 2017, Kamarudin et al. (DOI 10.1186/s12874-017-0332-6), identified 18 estimation methods of time-dependent ROC curve analyses for censored event times and three other methods can only deal with non-censored event times. Despite the considerable numbers of estimation methods, the authors concluded that the applications of the methodology in clinical studies are still lacking. This point was added to the discussion as well as the reference.

  1. Detailed inclusion and exclusion criteria are needed.

Answer: All consecutive patients aged 18 years or older surgically treated with curative intent for OSCC at the Instituto do Câncer do Estado de São Paulo, Hospital das Clínicas da Faculdade de Medicina da Universidade de São Paulo (ICESP HCFMUSP) from October 2009 to December 2018 were included. Patients with other previous oncological treatments, those with recurrent disease or with distant metastasis at presentation (M1), and those with SCC of the lip were not included. The text was rewrote to make these points more clear at the first sentence of the methods session.

Reviewer 2 Report

Comments and Suggestions for Authors

Dear authors,

Abstract 

The cohort comprised 600 patients, with 73,5% being male subjects (Table 1 resource)

Lines 36 to 38; 135, 137, 178, 199 - Please write the statistical P in italics

Results

Table 1 - Do the authors consider that the location of the tumor and habits such as smoking are demographic characteristics?

The authors mostly only refer to the T aspect of the TNM staging system. Is there any reason not to refer to N or M aspects?

Author Response

1. Abstract - The cohort comprised 600 patients, with 73,5% being male subjects (Table 1 resource); Lines 36 to 38; 135, 137, 178, 199 - Please write the statistical P in italics.

Answer: Thank you. The informations were updated.

2. Table 1 - Do the authors consider that the location of the tumor and habits such as smoking are demographic characteristics?

Answer: We changed the term “demographic” for “patient’s / tumor’s characteristics”

3. The authors mostly only refer to the T aspect of the TNM staging system. Is there any reason not to refer to N or M aspects?

Answer: We included pN classification at the Table 1. M1 disease at the presentation was an exclusion criterion and we included this information on method’s session.

Reviewer 3 Report

Comments and Suggestions for Authors

The study aimed to assess the prognostic impact of hematological indices in patients with OSCC.

The paper is innovative, but there are still shortcomings that need to be revised:

1.The lymph node metastasis in Table 1 requires detailed classification.

2.It is necessary to construct a nomogram using independent risk factors and calibrate it using an external cohort for verification.

Comments on the Quality of English Language

The quality of English can be understood by readers.

Author Response

General comments: The study aimed to assess the prognostic impact of hematological indices in patients with OSCC. The paper is innovative, but there are still shortcomings that need to be revised:

1.The lymph node metastasis in Table 1 requires detailed classification.

Answer: We included pN classification at the Table 1.

2. It is necessary to construct a nomogram using independent risk factors and calibrate it using an external cohort for verification.

Answer: Thank you for your observations. AS pointed, the study does not provide a validation analysis using an external cohort of patients, what limits the calibration of the predictive results and does not bring the opportunity to build a nomogram, for example, a significant limitation that should be addressed. This point was added to the end of the discussion.

Reviewer 4 Report

Comments and Suggestions for Authors

This manuscript is technically correct, providing important pieces of information for a commune and a significant concern nowadays, oral squamous cell carcinoma. The analysis of clinical parameters correlated with histopathological data, the factors associated with shorter survival, and the increased RDW in patients with bad prognoses, are important findings for this subject. The authors are encouraged to continue the study in the direction of RDW associated with shorter survival of the patients.

I think the article can be published after some fine corrections:

Line 21:

The variables utilized to determine appropriate treatment in this disease à The variables utilized to determine appropriate treatment for this disease

Line 124:

Hematological indices were obtained from the last laboratorial exams collected àlaboratory exams….

Line 157-158:

Add articles: .. with a mean of…with an average of…

Line 163:

…. It is an inappropriate index

Line 165:

patients' à patient's

Line 238:

ration à ratio

The correlation of the biochemical and histopathological results supports the scientific quality of the article. The manuscript can be published after some technical editing corrections also according to the rigors of the journal (reference list for example).

Comments on the Quality of English Language

-

Author Response

General comments: This manuscript is technically correct, providing important pieces of information for a commune and a significant concern nowadays, oral squamous cell carcinoma. The analysis of clinical parameters correlated with histopathological data, the factors associated with shorter survival, and the increased RDW in patients with bad prognoses, are important findings for this subject. The authors are encouraged to continue the study in the direction of RDW associated with shorter survival of the patients. I think the article can be published after some fine corrections:

1. Line 21: The variables utilized to determine appropriate treatment in this disease à The variables utilized to determine appropriate treatment for this disease; Line 124: Hematological indices were obtained from the last laboratorialexams collected à … laboratoryexams….; Line 157-158: Add articles: .. with amean of…with an average of…; Line 163: …. It is an inappropriate index…; Line 165: patients' à patient's; Line 238:  ration à ratio

Answer: All sentences and others were reviewed accordingly. Thank you.

2. The correlation of the biochemical and histopathological results supports the scientific quality of the article. The manuscript can be published after some technical editing corrections also according to the rigors of the journal (reference list for example).

Answer: Thank you.

Round 2

Reviewer 1 Report

Comments and Suggestions for Authors

The authors did not address all issues I proposed.

Comments on the Quality of English Language

Minor revision

Author Response

Sorry for the misunderstand. At yours previous review, four proposes were made. Firstly was mentioned that the manuscript lacks of novelty. To date, our study is the biggest one at the specific literature. Moreover, this is the first report that evaluated several Hematological indexes all together, demonstrating the relationship between them and the relevance of each one. This point is highlighted at the discussion session. Secondly, the suggested key words were removed from the new version of the manuscript as suggested. Thirdly, you suggested that ROC curve is not suited for time-related endpoints included the OS. We completely understand this point and opted to maintain this methodology to study the cutoff values once it is the most used tool to access this values in the literature and we also did a discussion about this point at the discussion. Finally, we detailed inclusion and exclusion criteria as requested at methods session. We, please, kindly ask the review to understand the new report mentioned “The authors did not address all issues I proposed.” We did addressed the four questions performed by the reviewer.